# Dynamic Performance of Partially Orifice Porous Aerostatic Thrust Bearing

**DOI:** 10.3390/mi12080989

**Published:** 2021-08-20

**Authors:** Muhammad Punhal Sahto, Wei Wang, Ali Nawaz Sanjrani, Chengxu Hao, Sadiq Ali Shah

**Affiliations:** 1School of Mechanical and Electrical Engineering, University of Electronic Science and Technology of China (UESTC), Chengdu 611713, China; punhal76@yahoo.com (M.P.S.); alinawaz.sanjrani@muetkhp.edu.pk (A.N.S.); 13402831920@163.com (C.H.); 2Department of Mechanical Engineering, The University of Lahore, Defence Road, Lahore 54000, Pakistan; 3Department of Mechanical Engineering, Mehran University of Engineering and Technology, Shaheed Zulfiqar Ali Bhutto Campus, Khairpur Mir’s 66020, Pakistan; ssssadiqalishah@gmail.com

**Keywords:** computational fluid dynamic study, partial porous orifice, dynamic stiffness and damping, fluid flow

## Abstract

The aerostatic thrust bearing’s performance under vibration brings certain changes in stiffness and stability, especially in the range of 100 to 10,000 Hz, and it is accompanied by significant increase in fluctuations due to the changes in frequency, and the size of the gas film damping. In this research work, an analysis is carried out to evaluate the impact of throttling characteristics of small size orifice on stiffness and stability optimization of aerostatic thrust bearings. There are two types of thrust bearing orifices such as: partial porous multiple orifice and porous thrust bearings and their effects on variations in damping and dynamic stiffness are evaluated. A simulation based analysis is carried out with the help of the perturbation analysis model of an aerostatic thrust bearing simulation by using FLUENT software (CFD). Therefore, two models of aerostatic thrust bearings—one with the porous and other with partial porous orifice are developed—are simulated to evaluate the effects of perturbation frequencies on the damping and dynamic stiffness. The results reveal a decrease in the amplitude of dynamics capacity with an increase in its frequency, as well as a decrease in the damping of partial porous aerostatic thrust bearings with an increase in the number of orifices. It also reveals an increase in the radius of an orifice with an increment of damping of bearing at the same perturbation frequency and, with an increase in orifice height, a corresponding decrease in the damping characteristics of bearings and in the dynamic stiffness and coefficient of damping of bearing film in the frequency range less than 100 Hz.

## 1. Introduction

Aerostatic bearings have certain characteristics such as minimum friction, rotational accuracy, and ultra precision. Considerable research has been conducted on the performance of aerostatic thrust bearing in a bid to make certain improvements in its existing performance to meet their demands of improved performance in related industries such as Microelectronic, Defense, Semiconductors, Measuring instruments, Aerospace, and Textile. There are two main functions of aerostatic bearings such as reduction of friction and motion errors. Earlier studies have covered main parameters of aerostatic thrust bearings such as stiffness, load-carrying capacity, and static characteristics. However, the geometrical structure of aerostatic thrust is also an important parameter, which may bring certain positive changes in its performance. The bearings with an existence of the air film are capable of running without the friction and achieving greater accuracy of motion [1,2,3]. The distribution pressure of the film can be improved by bringing certain changes in its structure. Therefore, the revised form of thrust bearing is analyzed, and its performance is compared with a traditional orifice restrictor aerostatic bearing. Several studies were carried out based on improvement of the load capacity, stability, and stiffness, on the basis of proper partitioning of random holes for pressure distribution [4,5]. In this regard, a multi grid coupling dynamics model of 5 DOF was proposed for the spindle of aerostatic. The Thomas and ADI method based on a solution of the Reynold Equation was used along with the restoring force to resolve the problems of the fluctuating air film force with times [6]. Certain numerical and experimental studies have also been conducted since the inception of axial thrust bearings keeping in view its effects on performance on various load capacities [7]. A one-dimensional aerostatic model for circular porous air bearing was developed for onward simulation on the basis of Darcy’s flow theory. Computational methods were used with the objectives to optimize both the time and cost parameters [8]. In the proposed study, the equilibrium stability approach was considered, in order to reduce the instability of thrust bearing by the application of damping support. It is pertinent to mention that research was carried out in this regard revealing that the dynamics of thrust bearing improved with the help of flexible damping support on the basis of the comparison of the stiffness of the gas film [9]. In another study, it was concluded that, with the inclusion of pocket and central feed holes into the circular aerostatic thrust bearing, which were supported by rubber O rings, and with an arrangement of the fixed pad, its stability could be enhanced [10]. The Routh–Hurwitz criteria was also developed to estimate the rotor-bearing’s stability for optimization of stability [11]. Likewise, Nyquist criteria were applied to investigate the stability of externally pressurized bearings [12]. Certain studies were carried out on high speed bearing operations, which concluded that, with an increase in temperature at high speed, there is a failure of bearing stability and deterioration of the film of fluid [13]. The effects of static characteristics of the porous, single orifice, and multiple orifice types on the performance of aerostatic thrust bearings were also investigated [14]. Dynamic stability studies with spiral grooved thrust bearing were carried out by utilizing two methods. The first method was related with stability of film, and the second method provided a relationship of speed limit of bearing with unconditional stability [15]. Several studies were also conducted to determine the coefficients of the fluid film of a dynamic rotor using a hybrid bearing with various angled injections [16]. The Nyquist’s criteria based studies revealed certain limits of stability and its results reflected a relationship between the air gap and supporting structure [17]. Studies were also carried out to analyze the impact of changes in the geometrical shape of the bearing holes on its performance. In such a study, two types of bearings were proposed: one with a hole and the other with an annular groove supply to avoid deflection over the surface of aerostatic porous bearings [18]. Certain investigations were related to the effects of manufacturing errors on the load capacity of bearing and significant exaggeration of static performance due to the changes in the distribution of pressure and thickness of film [19]. Unique procedures and methods were used for the evaluation of effects of pressure depression such as; separation of variables and the boundary-layer equations method. These methods facilitated the evaluation of variable parameters, such as influences of orifice diameter, pressure supply, pressure ratio, and film thickness on the depression of pressure. In such a study, it was concluded that the depression of pressure lessens with reduced film thickness, pressure supply, and enhanced diameter of orifice [20]. Certain studies were based on the optimized algorithm to estimate an equivalent coefficient of damping for the system and simulated transient response of the displacement of the shaft and matched with measured shaft displacement [21]. Other studies were carried out to investigate the impact of the thickness of the material, Young’s modulus, and supply of air on the material of porous, and concluded with suggestions of a porous restrictor with a double layer [22]. The Finite Element Method (FEM) and Proportional Division Method (PDM) were proposed in certain research works for computation of an angular stiffness [19]. In another research work, a novel method of modeling for determination of the FSI spindle systems of aerostatic was suggested on the basis of the changes in the structure of air film. A virtual method of weight loading was also projected for the evaluation of the stiffness of the spindle by considering the gravitational eccentricity and deformation of structure, and its reliability was validated through test and trial experiments. The FSI model suggested aerostatic spindle effects of geometrical parameters such as air, film, and structure dimension on performance optimization [23]. The Active Dynamic Mesh Method (ADMM) was also proposed for the investigation of dynamical performance of a double-pad bearing according to the perturbation model. In another work, a model was proposed to investigate the effects of orifice diameter, supply pressure, eccentricity ratio, squeeze number, and the step reply of a double-pad bearing and was followed by its comparison with the Passive Dynamic Mesh Method (PDMM) [24]. The effects of changes in perturbation frequency on the coefficient of damping and dynamic stiffness with EEPG were also investigated [25]. The mathematical calculation method of partially porous aerostatic thrust bearings was carried out by using the Finite Element method [26]. Likewise, CFD Fluent software was used to investigate the aerostatic bearing’s working performance in the low vacuum condition [27]. The proposed aerostatic thrust bearing model, with a variable pressure-equalizing groove, revealed a different structure and certain increase in the bearing’s stiffness [28]. The use of microgrooves in the double row symmetric radial bearing with the combined external throttling system may decrease the circumferential flow of compressed air and, on the other hand, it will improve the capacity of bearing [29]. Simulation software computational analysis methods, such as the Computational Fluid Dynamics (CFD) method, Finite Element Method (FEM), and Finite Difference Method (FDM), are also used for analysis and calculation of damping and dynamic stiffness on the basis of the solution of partial differential equations. The simulation software provides efficient means for analysis of the effects of certain variables on the performance. However, there are certain demerits of the analytical calculation method, which are related to its accuracy, as the results accuracy is subject to comparing the subsequent results obtained through FDM and FEM, with similar analytical approaches. There are certain complications with the results obtained through these software programs, which are related to the quality of the mesh, convergence criteria, and coordination conditions. This research work provides two types of partial multiple orifice porous aerostatic thrust bearing models with an objective to obtain variations in damping and dynamic stiffness.

## 2. Methodologies

Simulation software based methods and experimental methods are proposed to evaluate the effects of changes in the variables and its impacts on the performance of aerostatic bearings. Details of such research methods are mentioned in the preceding sections, which are given below.

### 2.1. CFD Base Dynamic Grid Calculation Method

The CFD method is used in the research work to investigate the aerostatic bearing’s characteristics. The equation of Navier–Stokes (N–S) is solved by the CFD method by using the procedure of the 3D transient physical field of axial thrust bearing such as temperature, pressure, and velocity field. Two following types of porous and partial multiple orifice porous thrust bearings are used in this calculation method.

### 2.2. Fluid Control Equation

In the range of Newtonian fluid, the conservation laws of mass, momentum, and energy can be calculated by the N–S equation. The mass conservation law also called continuity equation is as below [30]:(1)$$\frac{\partial \rho }{\partial t}+\frac{\partial (\rho u)}{\partial x}+\frac{\partial (\rho v)}{\partial y}+\frac{\partial (\rho w)}{\partial z}=0$$

It and can also be written as:(2)$$\frac{\partial \rho }{\partial t}+div\left(\rho \vec{u}\right)=0$$
where ρ represents density, *t* represents time, and the velocity vector components of Cartesian coordinates *x*, *y*, and *z* are *u*, *v*, and *w*, respectively. The microelement is calculated by the sum of all forces and equals the change in momentum of fluid microelements, which is indicated by conservation momentum law and also the second law of Newton. The conservation momentum equation is given below [30], and it can be derived in three directions:(3)$$\left\{\begin{array}{c} \frac{\partial (\rho u)}{\partial t}+\mathrm{div}\left(\rho \bar{u}u\right)=-\frac{\partial \rho }{\partial x}+\frac{\partial \tau_{\mathrm{xx}}}{\partial x}+\frac{\partial \tau_{\mathrm{yx}}}{\partial y}\frac{\partial \tau_{xx}}{\partial z}+F_{x}\\ \frac{\partial (\rho u)}{\partial t}+\mathrm{div}\left(\rho \vec{uv}\right)=-\frac{\partial \rho }{\partial y}+\frac{\partial \tau_{xy}}{\partial x}+\frac{\partial \tau_{y}}{\partial y}\frac{\partial \tau_{y}}{\partial z}+F_{y}\\ \frac{\partial (\rho u)}{\partial t}+\mathrm{div}\left(\rho \vec{uw}\right)=-\frac{\partial \rho }{\partial z}+\frac{\partial \tau_{xz}}{\partial x}+\frac{\partial \tau_{y=}}{\partial y}\frac{\partial \tau_{m}}{\partial z}+F_{z} \end{array}\right.$$
where *P* is pressure of film, $\tau_{xx}$, $\tau_{xy}$, and $\tau_{xz}$ are the viscous stress components of fluid, $F_{x}$ is *x*-directional force, $F_{y}$ is *y*-directional force on body, and $F_{z}$ is force on the body in the *z*-direction. In this calculation method, since the physical force is due to gravity, and the *z*-axis is vertical upward; therefore, $F_{x}=0$, $F_{y}=0$, and $F_{z}=-\rho g$. According to energy conservation law, an increased energy rate in the microelement is equal to the net heat flow into microelements plus work done by surface force and physical force on the microelements, which is the thermodynamic first law and is indicated in the following form of equation [30]:(4)$$\frac{\partial (\rho T)}{\partial t}+\mathrm{div}\left(\rho \bar{ı}T\right)=\mathrm{div}(\frac{k}{c_{p}}\mathrm{grad}T)+S_{T}$$
where grad *T* is the temperature gradient, *k* is heat transfer co-efficient of fluid, $c_{p}$ is the specific heat capacity, and $S_{T}$ is viscous dissipation energy [30].
(5)$$\frac{\partial (\rho \phi )}{\partial t}+\mathrm{div}\left(\rho \vec{u}\phi \right)=\mathrm{div}\left(\Gamma \mathrm{grad}\phi \right)+S$$

It can become to following form [30]:(6)$$\begin{array}{c} \frac{\partial (\rho \phi )}{\partial t}+\frac{\partial (\rho u\phi )}{\partial x}+\frac{\partial (\rho v\phi )}{\partial y}+\frac{\partial (\rho w\phi )}{\partial z}\\ =\frac{\partial }{\partial x}(\Gamma \frac{\partial \phi }{\partial x})+\frac{\partial }{\partial y}(\Gamma \frac{\partial \phi }{\partial y})+\frac{\partial }{\partial z}(\Gamma \frac{\partial \phi }{\partial z})+S \end{array}$$
where ϕ is the general variable, *u*, *v*, and *w* are velocity components, *T* is the temperature, Γ is the co-efficient of diffusion, and *S* is the source term. For the aerostatic thrust bearing with orifices, the pressure distribution in the air gap can be achieved by solving Equation (Equation 6). However, as the air passes through the porous material, it is accompanied by a loss in viscosity, inertia, and velocity slip in the boundary layer in between the porous material and air gap. Thus, it is necessary to establish an internal fluid lubrication equation of porous material including viscosity loss, inertia loss, and velocity slip, based on accurate measurement of the characteristic parameters of porous material. The equation of internal fluid lubrication of porous material takes the following final form:(7)$$\nabla P_{i}=-\left(\sum_{j=1}^{3}D_{ij}uv_{j}+\sum_{j=1}^{3}C_{ij}\frac{1}{2}\rho \left|v\right|v_{j}\right)$$
(8)$$\frac{\partial (\gamma \rho )}{\partial t}+\nabla \left(\gamma \rho \vec{v}\right)=0$$
(9)$$\frac{\partial (\gamma \rho \bar{v})}{\partial t}+\nabla \left(\gamma \rho \vec{v}\times \vec{v}\right)=-\gamma \nabla I$$
where $D_{ij}$ expresses the matrix co-efficient related to the viscous permeability co-efficient, $C_{ij}$ shows a co-efficient matrix related to the inertia permeability co-efficient, and γ represents the porosity of porous material. The pressure distribution of porous aerostatic thrust bearing can be attained by solving the equation of fluid lubrication and the N–S equation. Likewise, the distribution of pressure in the air gap can also be attained. The methods are used for the computation of the characteristics of flow field of aerostatic thrust bearings such as distribution of pressure in the air gap. Such calculation of FEM, FDM, and FVM are resolved by using the Reynolds equation. Since the equation of Reynolds is the partial differential equation, so that the calculation efficiency of FEM and FDM methods is high, the Three-Region Theory is applied to supplement the Reynolds equation, whereas a drop in pressure is still an issue to express and to be characterized by a slow recovery near the orifice; this is how it provides an imperfect definition of parameters of material property. FLUENT software is utilized in fluid calculation for solution of the transient 3D N–S equation, which is based on FVM, and it is capable of describing characteristic parameters of porous material. FLUENT software solves the pressure distribution characteristics of flow field inside an aerostatic bearing and, on this basis, the other characteristics of the aerostatic thrust bearing such as dynamic stiffness and damping are analyzed. The computer with configurations having Random Access Memory (RAM) of 112 GB, equipped with the graphics of a NVIDIA Quadro K620-GPU(GM107) processor was used to perform this simulation of an aerostatic thrust bearing.

### 2.3. Computational Models and Grids

The 3D model of the flow calculation domain of the partial orifice aerostatic thrust bearing is used with the same two throttling modes. The fluid domain of orifice throttling is divided into the orifice region and gas film region, as shown in Figure 1. The inlet and outlet pressure are set at the upper surface of an orifice and outer side of film thickness, respectively. The partial porous material is set as porous walls. The wall surface under the gas film region and the partial region is set as wall2, and the region of porous is defined as porous and other surfaces like walls. The fluid calculation domain of partial porous throttling mode is similar to the small orifices, as shown in Figure 1. The throttling area includes the viscous permeability co-efficient, inertial permeability co-efficient, and porosity at the same time, the boundary layers are required to be set between the porous region and the gas film region. This ensures the calculation accuracy of the grids(mesh) of the two throttling areas in the direction of film thickness, and the grids of partial porous in the longitudinal direction are set up to 10 layers and 100 layers, respectively. The modeling, meshing, and geometric characteristics of aerostatic thrust bearing are defined with their differences and similarities which are compared with each other. The design parameters that define the boundary conditions and dimensions of modeling are given in Table 1.

The grids are determined through a sweep method by setting the 100 cell numbers for the orifices, and the 1000 numbers are fixed for porous thrust bearing. The hexahedral mesh element type is used, and the size of the mesh is fixed by default. The number of nodes and elements are given in Table 2. The effect of grids on the results of simulation are shown in Table 2 under supply pressure of 0.5 MPa, and the number of elements and nodes are 283,525 and 312,839 respectively. The X, Y, and Z directional lengths are 80 mm, 80 mm, and 8 mm, respectively, with the volume of 2560.4
${\mathrm{mm}}^{3}$. The parameters of numerical convergence in a fluid domain are set as different iteration numbers to attain the results of simulation. However, it was observed that the different iterations are not affecting the results. Therefore, the same iteration numbers are used to save time such as $1\times 10^{-6}$ in the residuals of continuity of *x*, *y*, and *z* velocity.

### 2.4. Dynamic Grid Method

This method is used for the calculation of static characteristics of aerostatic thrust bearing. The boundary of the fluid domain is fixed, but the grids and nodes are not fixed. The fluid control equation is discretized in the static coordinate system. To analyze the characteristics such as damping and dynamic stiffness of aerostatic bearings, the gas film bearing wall needs to move according to a given perturbation equation because the mesh in the fluid domain changes with time during the calculation of the transient flow field. Three equations of conservation of air film can be stated by Equation (Equation 6). The Finite Volume Method (FVM) is used by fluent in discretizing the equation of control. Three basic governing equations in the bearing gas film are expressed by the governing equation in mathematics. The fluid equations are then converted from volume integral to area integral by using a Gauss divergence algorithm. These equations are expressed as:(10)$$\begin{array}{c} \frac{d}{dt}\int_{V}\rho \phi dV+\int_{\partial V}\rho \phi \left(u-u_{g}\right)dA\\ =\int_{\partial V}\Gamma \nabla \phi dA+\int_{V}S_{\phi }dV \end{array}$$
where $S_{\phi }$ is the scalar source term, $u_{g}$ is the moving velocity, *A* is the area of control volume boundary, and dv is the closed boundary control volume. The first-order differential is used to get in the term of time differential:(11)$$\frac{d}{dt}\int_{V}\rho \phi dV=\frac{{\left(\rho \phi V\right)}^{n+1}-{\left(\rho \phi V\right)}^{n}}{\Delta t}$$
where *n* and n+1 show the current and next time steps, respectively, and the relationship satisfies
(12)$$V^{n+1}=V^{n}+\frac{dV}{dt}\Delta t$$

The derivative of the control volume to time can be expressed as:(13)$$\frac{dV}{dt}=\int_{\partial V}u_{g}dV=\sum_{j}^{n_{f}}u_{g,j}\cdot A_{j}$$
where $n_{f}$ is the number of all boundary faces on the control volume *V*. By solving Equations (10)–(13), the dynamic motion effect of the boundary on the transient flow field can be attained. Dynamic and smooth mesh methods are used for updating of the meshes. To effectively control mesh quality, the mesh is dynamically updated in time steps by using the Dynamic Layering and Smoothing methods. The dynamic layering method is only applicable to structural grids. The number of grid layers can be increased or decreased on the basis of the change of the height of the grid layer of the moving boundary, due to the large displacement amplitude of the moving boundary. In the smooth methods, mesh number does not change, and mesh boundary is balanced by the spring system, which is appropriate when the moving direction is perpendicular to the boundary, and the amplitude of moving boundary is small. Figure 2 shows the solution process flow chart of dynamic stiffness and damping coefficient of the porous aerostatic bearing.

## 3. Dynamic Stiffness and Damping Characteristics Analysis

The results are bifurcated in two categories such as those obtained through simulation and experimental methods. Details of parameter analysis are provided in the following sections.

### Dynamic Mechanical Properties of Gas Film

In this section, dynamic mechanical properties of gas film are analyzed, as is illustrated in Figure 3. The model of partial porous aerostatic is composed of partial orifices, gas film, and thrust pad, as is shown in Figure 3. In the modeling process, the outside surface of film thickness is defined as outlet pressure, and the upper surface of a partial porous orifice is defined as the pressure inlet. Fluent software is used for estimation of a cloud chart of the pressure distribution, the pressure value at any point, and numerical change curve. The cloud map of the pressure distribution was in the form of a straight line and the precise form of gas flow velocity vector. Finally, the distribution of pressure under the changed structural parameters such as supply of pressure of 0.5 MPa, the atmosphere pressure of 1.03 MPa, and remaining parameters are given in Table 1. The pressure distribution diagram along the radial at the exit of the partial porous orifice is shown in Figure 3, which includes a pressure profile from the center of an aerostatic thrust bearing radially to the periphery of the bearing. The film thickness is 10 μm; the diameter of orifice is 5 mm, and the permeability of porous material is $1\times 10^{-14}\;m^{2}$. The simulation results are obtained by Fluent software. Finally, a post-processing tool is used to process the results. The pressure clouds and numerical curves of the partial porous aerostatic bearings are obtained, which are shown in Figure 4. The pressure distribution cloud of partial porous aerostatic bearing is shown in Figure 4a having air supply pressure of 0.5 MPa, gas film thickness 10μm, permeability of $1\times 10^{-14}\;m^{2}$, the height of orifice of H=4 mm, the orifice radius of 1 mm, and the number of orifices N=4. The pressure variation curve of film gap along the *x*-axis under the same conditions is shown in Figure 4b. Figure 5 shows a velocity vector distribution of gas flow in the gas film formed from the pressure inlet of a partial porous orifice to the film gap. The velocity vector diagram of the whole bearing is shown in Figure 5a. It is clear from Figure 5b that the gas velocity reaches the maximum value at the pressure outlet; then, it gradually decreases and rises rapidly at the position of the partial porous orifice. When it is at the center of the bearing, the gas velocity drops rapidly close to zero because the gas forms a larger vacuum in the center position, making the pressure here reach the maximum value.

## 4. Experimental Setup

The experimental setup was used to evaluate the effects of variation in the parameters on the performance of axial thrust bearing as is shown in Figure 6. The setup includes an air compressor, vibration isolation platform, pressure control valve with pressure gauge, inlet pressure pipe, thrust bearing, sensors, air hose pipe, shell and rotor, simulation module, and a display device. The sole objective of this set up is to obtain the physical values of various parameters of thrust bearing such as displacement, velocity, and acceleration, to get the stability effects.

## 5. Analysis of Factors Influencing Damping, Static, and Dynamic Stiffness Characteristics

A simulation model is established on the basis of the structural parameters of porous aerostatic bearings. The static pressure distribution of the bearing is obtained by the application of the gas film steady lubrication theory under assumed conditions of small external disturbance. In order to evaluate the effects of an additional external disturbance on the transient flow field in the air film clearance, the level of disturbance is enhanced. Newton’s second law is applied to obtain the values of dynamic film pressure distribution, dynamic stiffness, and damping coefficient, and their values on the basis of occurrence of the changes in bearing displacement and bearing capacity. It is assumed in the calculations of dynamic stiffness and damping coefficient of a bearing film that, when the bearing is subject to a harmonic disturbance in the direction of the film, i.e., *z*-direction, the pressure of the bearing film changes with the changes in the level of external disturbance. If it is also assumed that the film thickness of air is $h_{0}$ of the bearing at equilibrium, the resultant change in the amplitude of the simple harmonic disturbance on the bearing surface is Δh and the frequency of the disturbance is ω, then the changes in the air film thickness with respect to time can be expressed by a relationship, shown in the following equation:(14)$$h=h_{0}+\Delta h$$

Since the dynamic film lubrication equation of porous aerostatic bearing for a compressible object was obtained in Section 2, the value of *h* given in Equation (Equation 14) is put into Equation (Equation 10), in order to obtain a correlation between Δp/Δh and ω, which is shown in Equation (Equation 15):(15)$$\begin{array}{cc} \; & 2\{\frac{\partial }{\partial x}[h_{0}^{3}\frac{\partial }{\partial x}(\frac{\Delta p}{\Delta h})]\}+\{\frac{\partial }{\partial y}[h_{0}^{3}\frac{\partial }{\partial y}(\frac{\Delta p}{\Delta h})]\}\\ \; & +3[\frac{\partial }{\partial x}(h_{0}^{2}\frac{\partial p_{0}^{2}}{\partial x})+\frac{\partial }{\partial y}(h_{0}^{2}\frac{\partial p_{0}^{2}}{\partial y})]=j24\mu (h_{0}\frac{\Delta p}{\Delta h}\omega +p_{0}\omega ) \end{array}$$

Equation (Equation 15) shows a nonlinear correlation between Δp/Δh and perturbed frequency ω. A further solution of Equation (Equation 15) leads to the complex number solution as is shown in the equation given below:(16)ΔpΔh=Rω,p0,h0+jImω,p0,h0
where *R* represents the real part and Im is the imaginary part in the above equation. Since the bearing capacity of a porous aerostatic thrust bearing is the integral sum of the air film pressure in the bearing gap on the entire surface region of the porous, Equation (Equation 16) is therefore solved further to obtain the values of the dynamic stiffness, which is given as follows:(17)K*=∫Rω,p0,h0dA+j∫Imω,p0,h0dA

The real part in Equation (Equation 17) is the dynamic stiffness *K* of the bearing, and the imaginary part is the product of the damping factor *C* and the disturbance frequency. These parts are indicated separately in the following equation:(18)K=Rω,p0,h0dAC=∫Imω,p0,h0/ωdA

The values of dynamic stiffness and damping coefficients of the gas film is reflected in Equation (Equation 18), which shows a relationship to the external disturbance frequency, the gas pressure before the disturbance, and the thickness of the air film. Therefore, the influence of the disturbance amplitude, film thickness, gas supply pressure, and permeability of porous materials on the dynamic stiffness and damping of the bearing are analyzed.

### 5.1. Influence of the Number of Partial Orifices (N = 4, 5, 6, 7)

The pressure distribution diagram of partial porous aerostatic bearing having air supply pressure of 0.5 MPa, gas film thickness of 10μm, permeability of $1\times 10^{-14}$$m^{2}$, the height of orifice *H* = 4 mm, the radius of orifice *r* of 1 mm, and the number of orifices *N* of 4,5,6,7 is shown in Figure 7. However, Figure 8 shows the changes in pressure distribution of partial porous aerostatic thrust bearing with different orifices’ numbers. It reveals that the number of orifices of partial porous aerostatic thrust bearing has an important impact on the pressure of bearing. It shows that, when there are four orifices, the pressure of bearing is 0.23 MPa, which increases to 0.278 MPa due to an increase in the number of orifices to 7, along with increase in the static bearing capacity of the partial porous aerostatic bearing up to a uniform 20.8%, and the uniform distribution of pressure on the surface of thrust plate of a bearing. A relationship between dynamic stiffness and damping curves of partial porous aerostatic thrust bearing at different orifice numbers is analyzed, which is reflected in Figure 9. It reveals that, in the low-frequency range (less than 100 Hz), the dynamic stiffness of the partial porous thrust bearing is constant, but it rises rapidly in the intermediate frequency range of $10^{3}$ to $10^{4}\;\mathrm{Hz}$, and then slows down gradually in the high-frequency range $10^{5}$ to $10^{6}$ Hz as is reflected in Figure 9a. Likewise, another relationship that exists between dynamic stiffness and perturbation frequencies is also analyzed, which reveals that the dynamic stiffness of bearing remains constant under the same perturbation frequency with an increase in the number of orifices. However, it reveals that the damping of partial porous aerostatic bearing gradually decreases with an increase in external perturbation frequency until its drop to zero, as is shown in Figure 9b. The dynamic damping curves of four kinds of partial porous orifices, in the low-frequency range (less than 100 Hz), indicates a gradual decrease in damping coefficient of partial porous bearing with subsequent increase in the number of orifices. Therefore, according to the comprehensive analysis, it can be concluded that the orifice number has a great influence on the static bearing capacity of the partial porous aerostatic thrust bearing but without influence on the bearing film’s dynamic stiffness. It can also be said that, with the increase in the number of orifices, the damping parameters of partial porous bearings become smaller, so the effect of increasing the number of orifices on the dynamic characteristics of bearings is not apparent.

### 5.2. Influence of Radius of Partial Orifice (r = 0.5 mm, 15 mm, 1.5 mm, 2 mm)

At this stage, the effects of changes in the radius of partial porous on pressure distribution are evaluated. These effects are reflected in Figure 10, following the setting of bearings’ gas supply pressure of 0.5 MPa, a gas film thickness of 10μm, a permeability of $1\times 10^{-14}\;m^{2}$, the height of orifice *H* of 4 mm, and number of orifices *N* of 4 at orifices radii, and *r* of 0.5 mm, 15 mm, 1.5 mm, 2 mm, respectively. The changes in the bearing pressures with different position are shown in Figure 11. The comparative analysis of the orifice radius of the partial porous aerostatic bearing leads to the proposition of a significant nonlinear increase of pressure distribution of the bearings with an increase of the radius. It also indicates proportional change between the radius of partial porous material and the pressure distribution of the bearings. When the radius *r* = 0.5 mm, the pressure of the bearing is 0.12 MPa, and when the radius *r* = 2 mm, the pressure of the bearing increases to 0.35 MPa. It also indicates that, when the radius increases, the distribution of pressure on the partially porous thrust surface is more uniform, which further improves the static bearing capacity and stiffness parameters of an aerostatic bearing. Based on the static carrying capacity of the above-mentioned partial porous orifice aerostatic thrust bearing, the dynamic simulation calculation establishes the influence of the orifice radius on the damping and dynamic stiffness of the film of the partial porous orifice aerostatic thrust bearing, which is depicted in Figure 12. Figure 12a shows that the external perturbation frequency increases, as dynamic stiffness of partial porous orifice bearing firstly decreases slowly, and then rises rapidly in the intermediate frequency range $10^{3}$ to $10^{4}$ Hz. However, it reveals that, in the high-frequency range $10^{5}$ to $10^{6}$ Hz, it gradually slows down, but it still shows a steady rise. Therefore, by comparing the dynamic stiffness curves of four different orifice radii, it can be concluded that the dynamic stiffness of bearing film increases with the increase of orifice radius at the same external perturbation frequency, but, with the rise of external perturbation frequency, the damping of the partial porous bearing film gradually decreases, and finally decreases to zero. It also indicates that, by comparing the dynamic damping parameters of four kinds of orifice radii, it can be concluded that, in the low-frequency range (less than 100 Hz), the bearing damping increases with the increase of orifice radius at the same perturbation frequency as shown in Figure 12b. Therefore, the analysis leads to the conclusion that the orifice radius of the partial porous orifice aerostatic thrust bearing has a great influence on the dynamic stiffness and damping of the bearing film.

### 5.3. Influence of the Height of the Partial Porous Orifice (H = 2 mm, 4 mm, 6 mm, 8 mm)

In this analysis, a relationship between pressure distribution of porous orifice aerostatic thrust bearing and different orifice heights is carried out. The pressure distribution of partial porous orifice aerostatic thrust bearing under different heights of the orifice is shown in Figure 13, at air supply pressure of 0.5 MPa, the film thickness of 10μm, permeability of $1\times 10^{-14}\;m^{2}$, the orifice radius *r* of 1 mm, and the number of partial porous orifices of 6 and the height of the orifices *H* of 2 mm, 4 mm, 6 mm, and 8 mm, respectively. The changes are reflected in Figure 14, which depicts the pressure distribution of a partial porous aerostatic bearing under different orifice heights. This analysis leads to the conclusion that the height of the orifice of the partial porous aerostatic bearing has a great impact on the pressure distribution of the bearing. It indicates that, when the height of the orifice is H=2 mm, the pressure of the bearing is 0.42 MPa, and, when the height of the orifice is 8 mm, the 0.265 MPa is the pressure of the bearing, and the bearing static carrying capacity drops by 58% with an increase of orifice height, which indicates that, with increasing orifice height, the pressure of the partial porous aerostatic bearing decreases, and, with the increase of the height of the orifice, there is a downward trend. The dynamic stiffness and damping of partial porous orifice aerostatic thrust bearing at different orifice heights are calculated using dynamic simulation, as is shown in Figure 15. According to these calculations, as the external perturbation frequency increases, the dynamic stiffness of a partial porous orifice bearing in the low frequency (less than 100 Hz) and medium frequency ($10^{3}$–$10^{4}$ Hz) remains constant initially and then it rises gradually in the high-frequency range ($10^{5}$–$10^{6}$ Hz), as illustrated in Figure 15a—and, likewise, the dynamic stiffness of the bearing under different orifice heights, as compared with the partial porous aerostatic bearing analyzed. According to these calculations with increasing height of an orifice, the dynamic stiffness of the bearing under the same disturbance frequency presents a significant decline, in which the height of the orifice is 8 mm, as is shown in Figure 15b. Therefore, it can be concluded that the damping of the partial porous aerostatic thrust bearing decreases gradually with the increase of the external perturbation frequency, and the final decline tends to zero. Therefore, it can be concluded that, when orifice heights increase, bearing damping characteristics in the low-frequency range (less than 100 Hz) decrease significantly. From the comprehensive analysis of Figure 15, it can be observed that the height of porous orifice has an obvious influence on the dynamic stiffness and damping of the partial porous aerostatic thrust bearing, as the dynamic stiffness and damping coefficient of bearing film decrease with the increase of orifice height.

## 6. Validation of Numerical Results through Experiments

Equation (Equation 16) verifies the results of dynamic stiffness numerically based on real and imaginary parts of the equation, whereas, in Equation (Equation 17), the two parts are further elaborated to determine the dynamic stiffness, damping factor, and the disturbance frequency. Equation (Equation 18) correlates the relationship with the above-mentioned equations by changing the variables such as disturbance amplitude, film thickness, gas supply pressure, and permeability of porous materials as analyzed for the calculation of damping and dynamic stiffness of the aerostatic bearing. Further changes in the time interval and its effects on acceleration and inlet pressure are analyzed at this stage of the research process. Initially, it was set for 10 periods with a frequency started from 3 s along with a constant increase in a time interval of 0.002 s. The interval changes, the acceleration changes, and the inlet pressure changes are shown in Figure 16. Since the reflected acceleration is constant, the vibration is stable and is controlled at the side of the shell. The rotor has little variation from the central limit (CL), and vibration displacement is around 2μm while the thickness of the gas film is about 20μm.

It can be inferred that the direction of *x* and *y* has the highest vibration and that the displacement amplitude is about 10 percent of the gas film. The acceleration, velocity, and displacement of bearing shell with the inlet pressure of 0.62 MPa, the amplitudes of $0.9\;m/s^{2}$ and $0.5\;m/s^{2}$ in the direction of *x*, and the amplitude in the *z*-direction, respectively, are illustrated in Figure 17. The thickness of the gas film was set as 20μm, and the vibration displacement as 0.5μm. It was observed that the amplitude of the shell was 20 times less than the amplitude of the rotor due to decrease in the effect of vibration on the shell. The changes in the amplitude of vibrations due to changes in inlet pressure are shown in Figure 18, which indicates different vibration conditions of the rotor in different directions. Accordingly, the pneumatic vibration occurred at 0.6 MPa of pressure, and the vibration started at maximum amplitude and at the inlet value of pressure 0.68 MPa, at 3.5 s, and it ended at 10 s. The only change in vibration of bearing was observed at the inlet pressure within the time span of 10 s. It was also observed that the amplitude changed with the value of inlet pressure, while the frequency was constant at 630 Hz. Certain changes in the amplitude of vibrations with inlet pressure were observed, which are illustrated in Figure 19, which indicates that, with the increase in inlet pressure, there is some increase in the amplitude of vibration. The vibration state of the shell was observed in three directions. It was noticed that a pneumatic type of vibration occurred at the pressure of 0.6 MPa, whereas the highest amplitude was observed at the 0.68 MPa pressure inlet value as the vibration started at 3.5 s, and it ended after a time span of 10 s. It was also observed that, despite increase in the inlet pressure value, there was no change in the value of frequency, which was 630 Hz. The amplitude of the shell was found to be lower than that of the rotor at the same frequency. In order to evaluate the effects on acceleration due to changes in the position of bearing, the bearing was set up at the vertical position, and the results of the rotor were obtained, which are illustrated in Figure 20a. It was observed that the acceleration was maximum in the direction of *y* and minimum in the direction of *x* at the inlet supply pressure of 0.62 MPa, while displacement of 0.25μm and the thickness of the film of 20 µm, respectively. It was also observed that the vibration of the rotor in a horizontal position was greater than the rotor put in a vertical position, as is shown in Figure 20b, as is visible in the appearance of increase in the acceleration amplitude of the rotor with a subsequent rise in inlet pressure. Therefore, the high amplitude was recorded at the inlet pressure from 0.62 MPa to 0.68 MPa along with the beginning of vibration at 3.5 s and its finishing at 10 s. The effects of changes in inlet pressures on acceleration amplitude were also analyzed at a later stage. It was observed that, at 624.2905 Hz frequency, amplitude changes with the increase of inlet pressure, as is reflected in Figure 21a, which indicates that the acceleration amplitude increases by 0.48 ${m/s}^{2}$ with the rise in inlet pressure. It was also observed that the displacement amplitude was maximum at 7.06 s, but it decreased when time increased to 7.068 s. The relationship between frequency of the shell and acceleration amplitude were further analyzed, which are illustrated in Figure 21b, and it shows that the acceleration amplitude was 0.089219 m/s${}^{2}$ at frequency of 624.2905 Hz. The vibration of the shell started from 3.5 s and ended at 10 s. It was also noticed that the vibration of the isolation platform was in xy directions, and the acceleration amplitude increased to 0.55 ${m/s}^{2}$ with an increase in pressure at the time of 7.052 s and reduced after 7.07 s. The effects of time span changes on the displacement amplitude were also analyzed. It was observed that the displacement amplitude was maximum at 7.05 s and minimum at 7.07 s, which is shown in Figure 22a. The vibration of the platform begins from 3.5 s and ends at 10 s as depicted in Figure 22b.

## 7. Conclusions

The simulated and experimental results reveal optimization of the dynamic stiffness and damping characteristics of the aerostatic porous thrust and partial porous orifice type bearings on account of changes in design and operating parameters such as orifice diameter, orifice height, material permeability coefficient, and orifice distribution. The experimental results of the dynamic stiffness and damping characteristics of the proposed parameters of a porous and partial porous aerostatic thrust bearing are compared with the simulated results to verify the accuracy of the simulation results on dynamic stiffness and damping characteristics. The dynamic stiffness and damping parameters reveal a nonlinear correlation between dynamic stiffness and damping characteristics at less than a 100 Hz frequency level. However, in the range of 100 Hz to 10,000 Hz, the amplitude of dynamic carrying capacity fluctuations shows a linear relationship. It indicates that, with a change in frequency, there is a change in the size of the gas film damping along with the variations in the dynamic bearing capacity of the gas film surface of the porous aerostatic thrust bearing in 10 cycles. Therefore, it can be concluded that, in the range of 100 Hz to 10,000 Hz with an increase in frequency, there is a significant decrease in amplitude of dynamic bearing capacity fluctuation, which indicates the influence of throttling on dynamic stiffness and damping characteristics of aerostatic thrust bearings film due to changes in the diameter of partial porous orifice aerostatic thrust bearing. It may also be concluded that a larger radius of partial porous aerostatic thrust bearing produces greater dynamic stiffness and damping of the partial porous aerostatic thrust bearing. It can also be concluded that a nonlinear relationship exists between orifice heights and damping characteristics of bearings at a frequency less than 100 Hz. 

## Figures and Tables

**Figure 1 micromachines-12-00989-f001:**
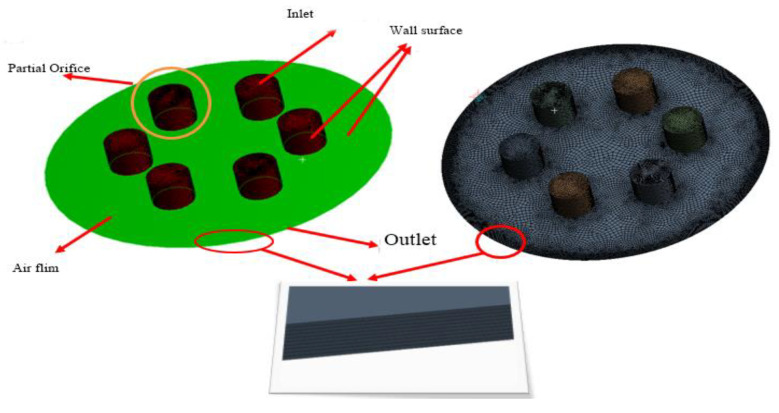
Partial porous orifice aerostatic thrust bearing model.

**Figure 2 micromachines-12-00989-f002:**
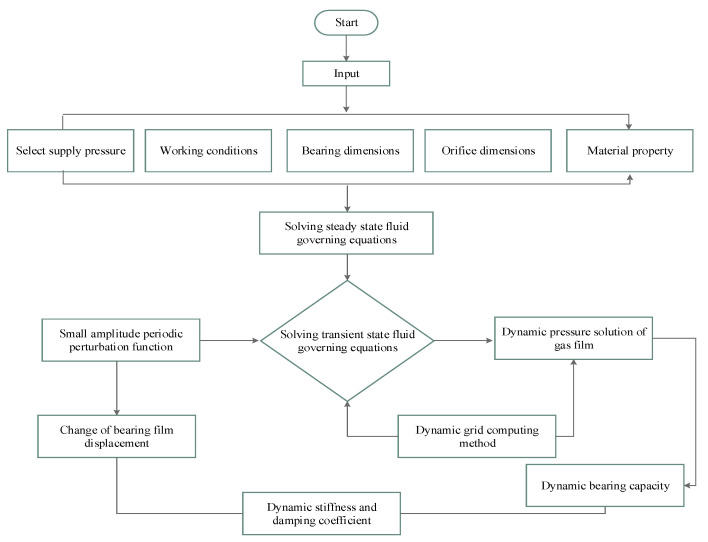
The flow chart determining the dynamic stiffness and damping coefficient.

**Figure 3 micromachines-12-00989-f003:**
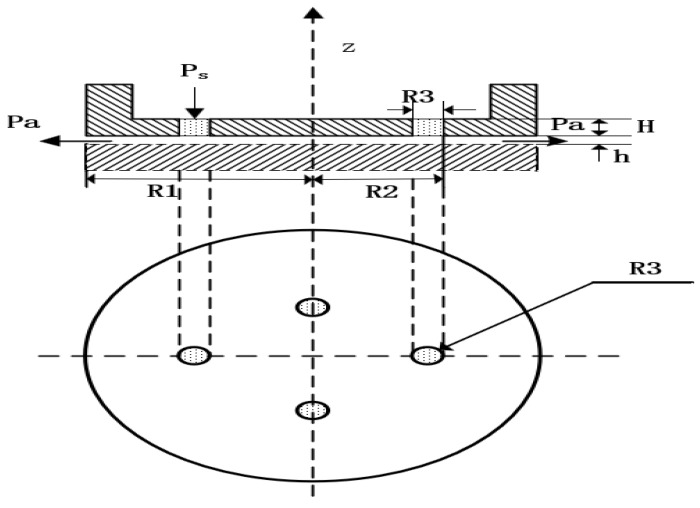
Diagram of partial porous orifice aerostatic thrust bearing.

**Figure 4 micromachines-12-00989-f004:**
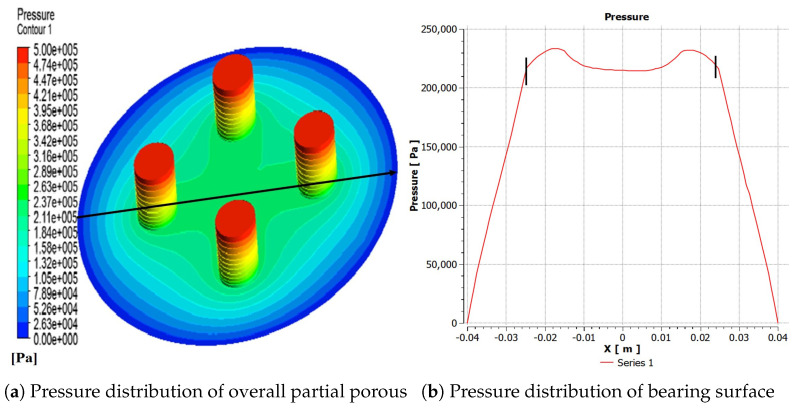
Partial porous orifice pressure distribution.

**Figure 5 micromachines-12-00989-f005:**
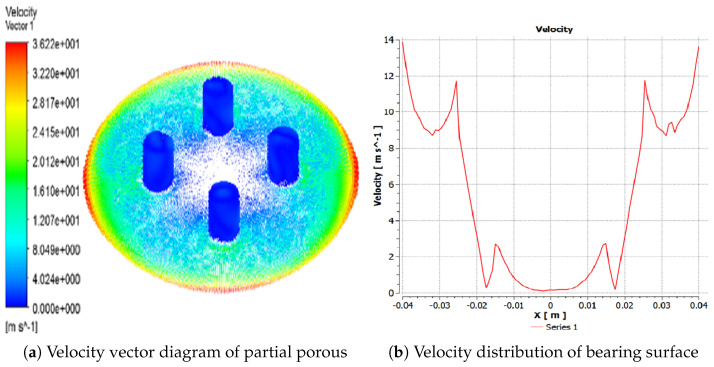
Partial porous orifice velocity distribution.

**Figure 6 micromachines-12-00989-f006:**
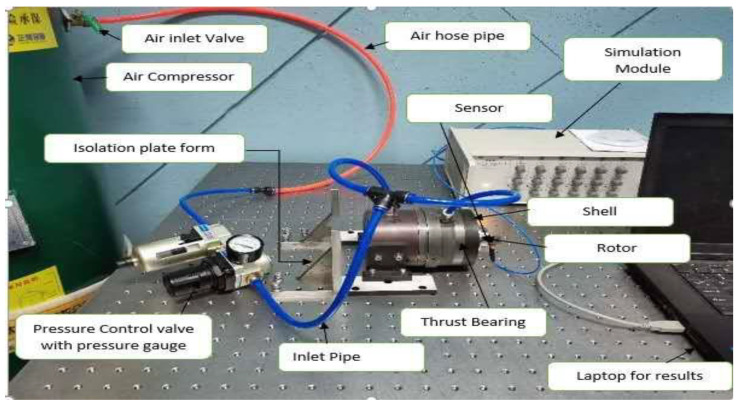
Experimental setups for aerostatic thrust baring.

**Figure 7 micromachines-12-00989-f007:**
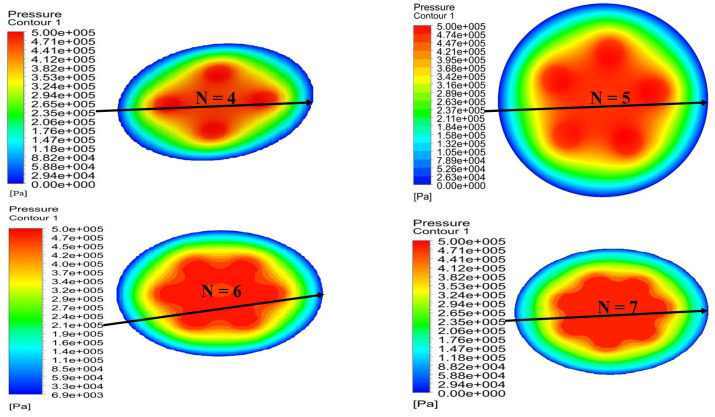
Cloud map of bearing pressure distribution when the number of partial porous orifices is different.

**Figure 8 micromachines-12-00989-f008:**
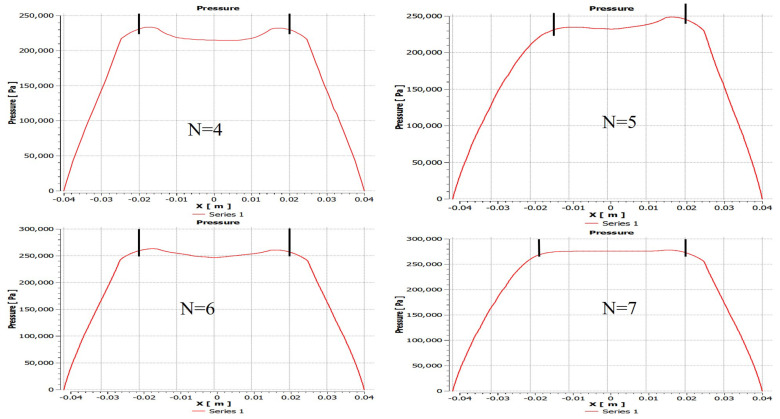
Distribution of bearing pressure with a different number of partial porous orifice.

**Figure 9 micromachines-12-00989-f009:**
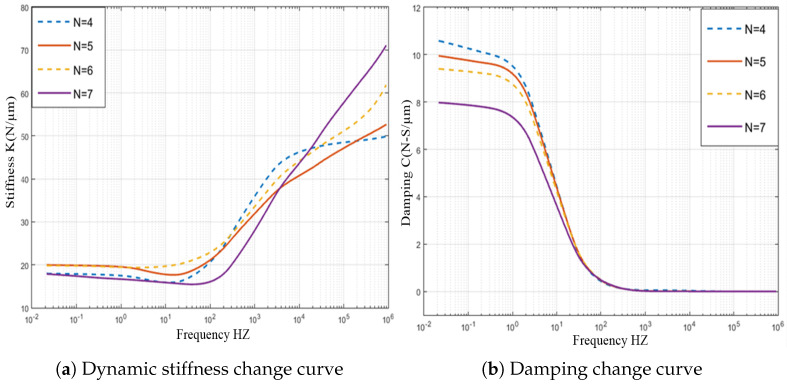
Dynamic stiffness and damping curve of thrust bearing with the different number of partial porous orifices.

**Figure 10 micromachines-12-00989-f010:**
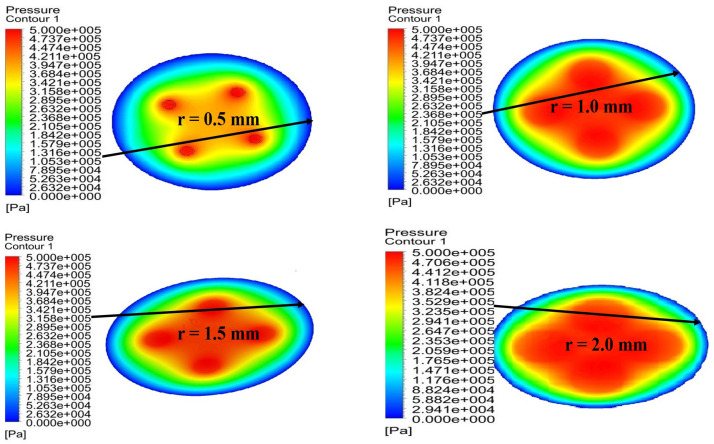
Cloud chart of bearing pressure distribution with a different radius of partial porous orifice.

**Figure 11 micromachines-12-00989-f011:**
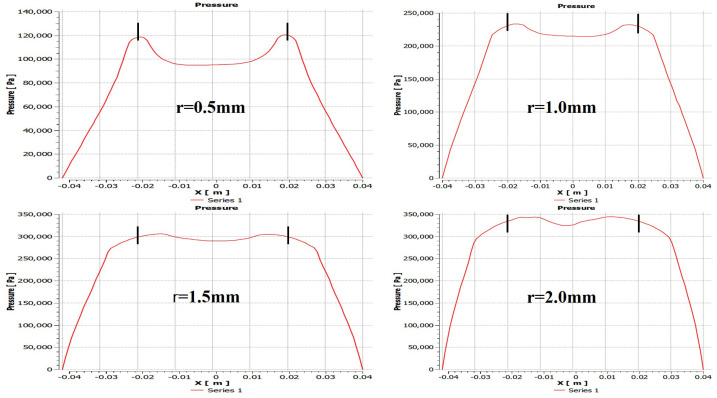
Distribution of bearing pressure with bearing when the radius of the partial porous orifice is different.

**Figure 12 micromachines-12-00989-f012:**
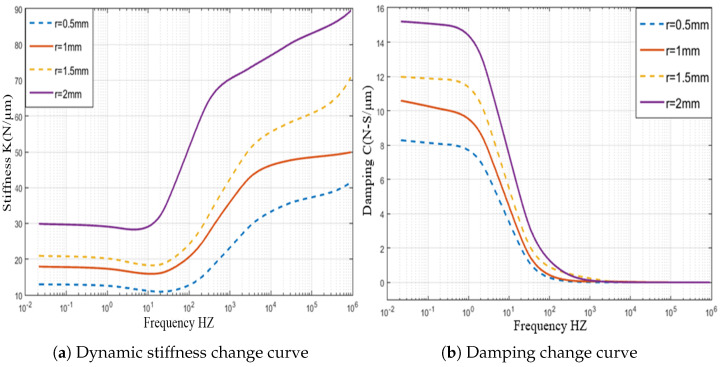
Dynamic stiffness and damping curve of thrust bearing with a different radius of partial porous orifices.

**Figure 13 micromachines-12-00989-f013:**
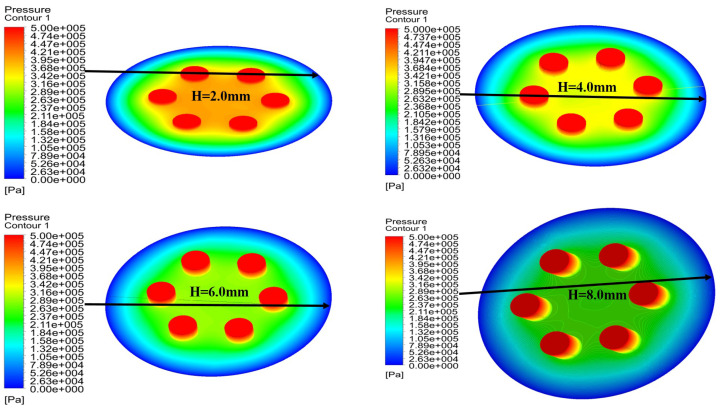
Cloud distribution pressure of the bearing with different height of the partial porous orifice.

**Figure 14 micromachines-12-00989-f014:**
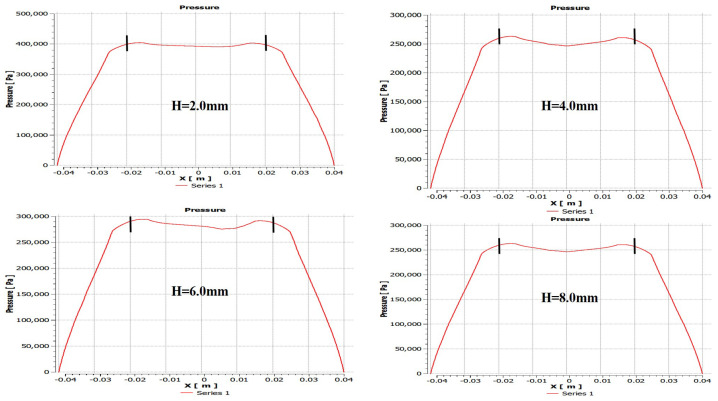
Bearing pressure distribution with bearing when the height of the partial porous orifice is different.

**Figure 15 micromachines-12-00989-f015:**
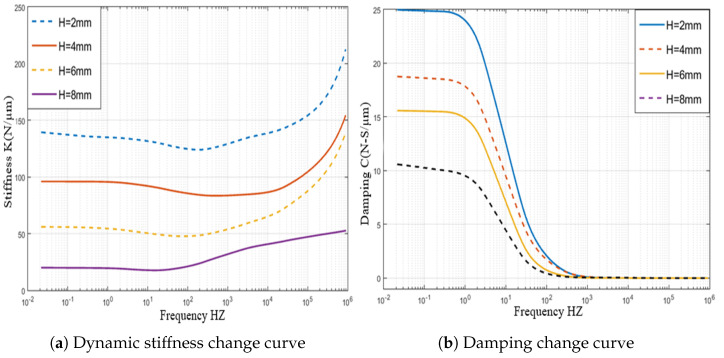
Dynamic stiffness and damping curve of thrust bearing with different height of partial porous orifice.

**Figure 16 micromachines-12-00989-f016:**
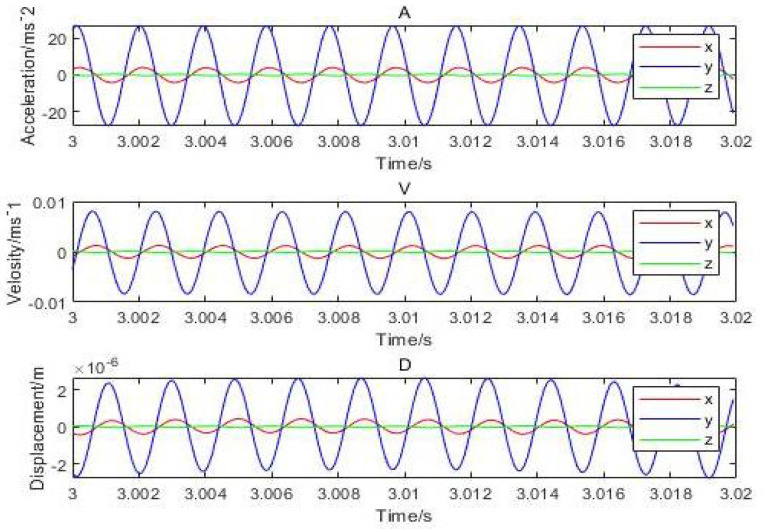
Dynamic acceleration, velocity, and displacement of rotor thrust bearing with (630 Hz) perturbation frequencies (*h* = 20 μm and Ps = 0.62 MPa).

**Figure 17 micromachines-12-00989-f017:**
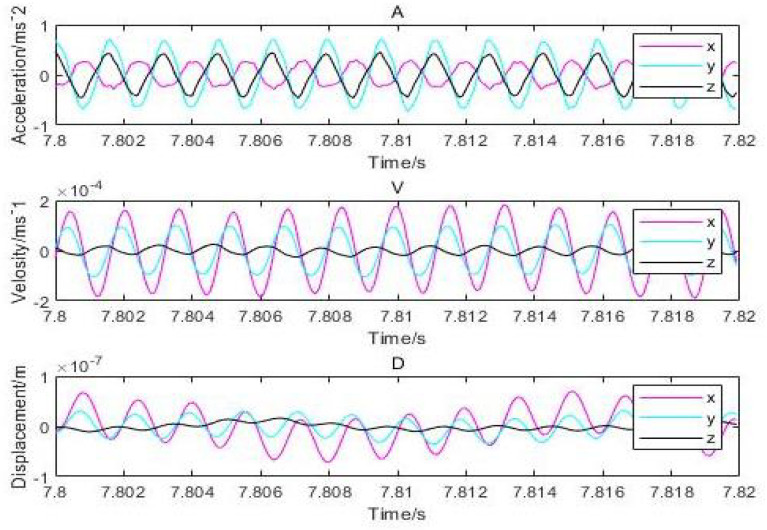
Dynamic acceleration, velocity, and displacement of shell bearing with (630 Hz) perturbation frequencies: (*h* = 20 μm and Ps=0.62 MPa).

**Figure 18 micromachines-12-00989-f018:**
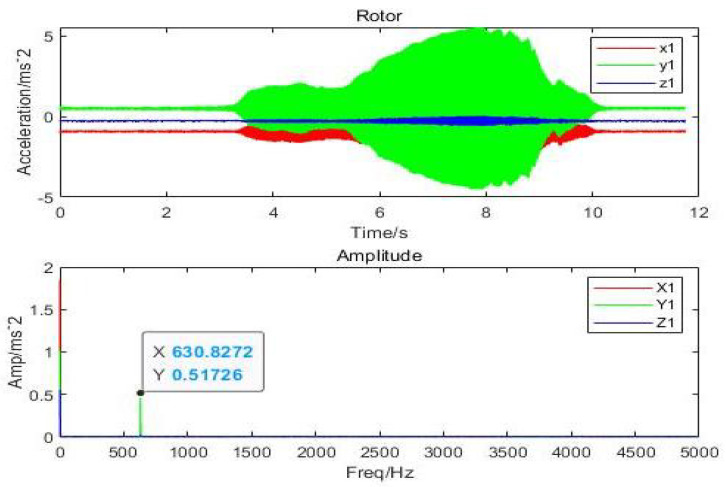
Amplitude of rotor of aerostatic thrust bearing with (630.8272 Hz) perturbation frequencies: (h=20μm and Ps=0.62 MPa).

**Figure 19 micromachines-12-00989-f019:**
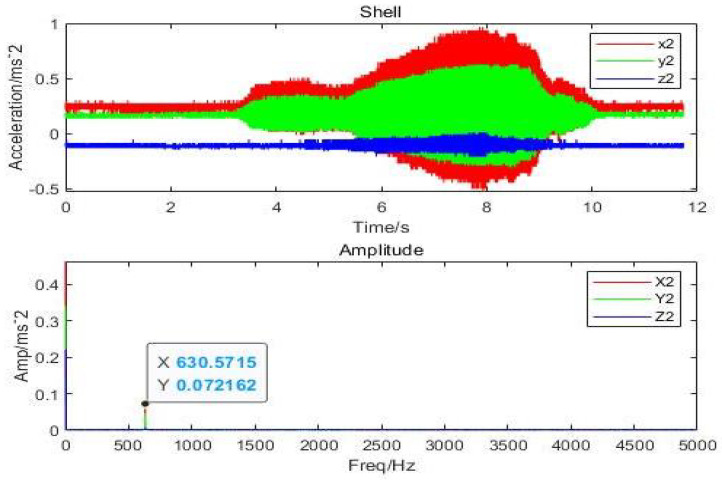
Amplitude response of shell of aerostatic thrust bearing with (630 Hz) perturbation frequencies: (h=20μm and Ps=0.62 MPa).

**Figure 20 micromachines-12-00989-f020:**
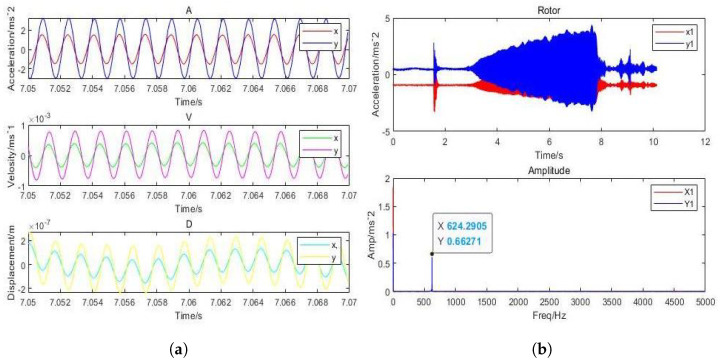
(**a**) Dynamic acceleration response of rotor of aerostatic thrust bearing with (630 Hz) perturbation frequencies; (**b**) amplitude response of rotor of aerostatic thrust bearing with (630 Hz) perturbation frequencies. (h=20μm and Ps = 0.62 MPa).

**Figure 21 micromachines-12-00989-f021:**
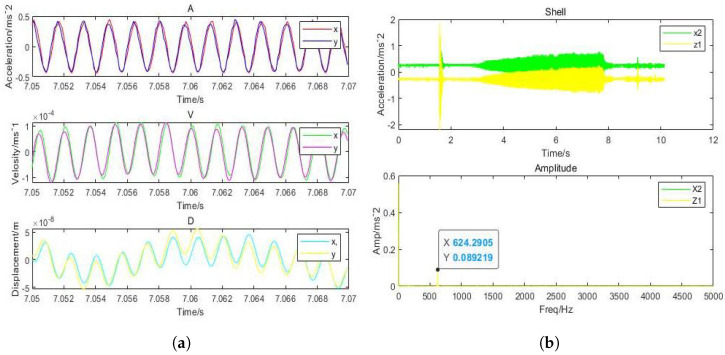
(**a**) Dynamic acceleration of shell with (630 Hz) perturbation frequencies; (**b**) amplitude of shell aerostatic thrust bearing with (630 Hz) perturbation frequencies: (h=20μm and Ps = 0.62 MPa).

**Figure 22 micromachines-12-00989-f022:**
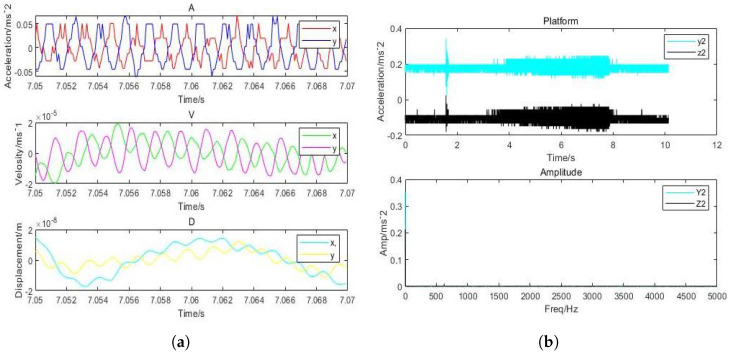
(**a**) dynamic acceleration of shell with (630 Hz) perturbation frequencies; (**b**) amplitude of shell aerostatic thrust bearing with (630 Hz) perturbation frequencies: (h=20μm and Ps = 0.62 MPa).

**Table 1 micromachines-12-00989-t001:** Design parameters and boundary conditions.

Design Variable	Orifice & Porous
Outlet pressure	0 bar
Inlet pressure	5 bar
Operating pressure	101,325 Pa
Temperature	293 K
Dynamic viscosity	$1.789\times 10^{-5}$ Ns/(m)
Gas density	$1.189\;{\mathrm{kg}/m}^{3}$
Film thickness	5–15 μm
Bearing outer diameter	80,000 μm
Bearing inner diameter	48,000 μm
Orifices height	2 mm/4 mm/6 mm/8 mm
Orifices radius	0.5 mm/1 mm/1.5 mm/2 mm
Viscous resistance coefficient	$1\times 10^{13}\;m^{-2}/1\times 10^{13}\;m^{-2}/1\times 10^{13}\;m^{-2}$
Porosity	0.3

**Table 2 micromachines-12-00989-t002:** Grid division information of aerostatic thrust bearing.

Object Name	Geometry
**State**	**Fully Defined**
**Bounding box**	
Length X	80 mm
Length Y	80 mm
Length Z	8 mm
**Properties**	
Volume	$2560.4\;{\mathrm{mm}}^{3}$
Scale Factor Value	1
**Statistics**	
Bodies	9
Active Bodies	9
Nodes	312,839
Elements	283,525
Mesh Metric	None

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
