# Peer review of "Dynamic Performance of Partially Orifice Porous Aerostatic Thrust Bearing"

_micromachines, 2021, doi:10.3390/mi12080989_

Round 1

Reviewer 1 Report

From the reviewer’s point of view, the paper topic is interesting and it has been written in a good order. The Reviewer recommends a revision.

However, please address the following corrections in the revised version:

  1. Please stress the novelty of the work and its contribution to the state of the art with a clearer statement;
  2. Please summarize the introductory section by emphasizing the most relevant literature on the numerical modelling for instance;
  3. In terms of equations, could you please refer them to a reference, in this case the readers could refer to find out more details;
  4. Regarding the FE mesh, please provide some details on the number of nodes and elements;
  5. Have you made any convergence studies on the FE mesh densities? If so, how the results were affected?
  6. Figures must be reproduced with a higher quality;
  7. The legend bar must be reproduced in a better manner; for instance, in Figures, 3, 4 and 6, it is better to rebuild it, resize it. Do not STRETCH them;
  8. Please respect all the decimals; unify decimals specially in the tables;
  9. Conclusions shall be rewritten in a more comprehensive manner.

Very best

The Reviewer

Author Response

Responses to Reviewer 1 Comments

Manuscript ID: micromachines-1303474

Title: “Dynamic Performance of Partially Orifice Porous Aerostatic Thrust Bearing

The authors would like to thank the anonymous reviewers and the associate editor for their effort in reviewing the manuscript and for their valuable and constructive comments and fruitful observations that helped in improving the quality of the manuscript to a publishable standard. Detailed below are some responses to the reviewers’ comments and suggestions. Reviewers’ questions and comments are in BLACK and the authors’ answers and comments are in RED.

Point 1: Please stress the novelty of the work and its contribution to the state of the art with a clearer statement.

Response 1: The novelty of work is related with the improvement in the performance of aerostaatic thrust bearing due to resultant increase in stability of bearing by changes in the number orifices, orifices height and low range of frequency as reflected on page 1-4 and on 19-20.

Point 2: Please summarize the introductory section by emphasizing the most relevant literature on the numerical modelling for instance.

Response 2: The introductory section includes related and most relevant literature on numerical modelling.

Point 3: In terms of equations, could you please refer them to a reference, in this case the readers could refer to find out more details

Response 3: Relevant reference [29] for the reader’s review in more details is given on page 4-5.

Point 4: Regarding the FE mesh, please provide some details on the number of nodes and elements.

Response 4: FE mesh details regarding number of nodes and elements are provided separately in Table 3 in methodologies section on page 6-7.

Point 5: Have you made any convergence studies on the FE mesh densities? If so, how the results were affected?

Response 5: In the research work it is presumed that if number of elements and nodes are increased that will increase densities of mesh and on the basis to realize improved convergence.

Point 6: The legend bar must be reproduced in a better manner; for instance, in Figures, 3, 4 and 6, it is better to rebuild it, resize it. Do not STRETCH them.

Response 6: The legend bar is reproduced of Figures, 3, 4 and 6, and the sequence number of said Figures are changed as 4, 5, and 7 respectively. Please refer the updated Figures on the page number of 9 and 11.

Point 7: Please respect all the decimals; unify decimals specially in the tables.

Response 7: All the decimals are updated specially in the tables. Please refer to page number 6-7.

Point 8: Conclusions shall be rewritten in a more comprehensive manner.

Response 8: Conclusion is updated and rewritten in a comprehensive manner.

The authors hope that the revised manuscript has satisfied the requirements from the reviewers, Associate Editor and the editor-in-chief. We would like to thank them for their very constructive comments, which are very insightful and beneficial for improving the quality of the manuscript.

Best regards,

Muhammad Punhal Sahto

Reviewer 2 Report

  1. There are numerous places in the text with English grammatical errors. The authors should be full checking for grammar and mistakes to meet the quality of Journal.

  1. 2. Most equations Tables in the paper, the authors don't mention to any references. Add citation to the equations which took from references.

  1. Add a new flowchart to explain the details of the selected approach.
  2. Which software/ program that used as a tool to solve the equations of your problem?

Give enough details about this point.

You mention that authors used FLUENT to find the solution. But there are many equations included in the paper. Did authors used all these equations?

Or just presented it.

  1. Did you check the accuracy of your numerical solution?

  1. How did you select the final numerical model?

Based on what, explain this point?

  1. 7. There are many research papers study the same problem that investigated in the present paper. What is exactly the new point of this work?

The authors should focus to clarify this issue in the paper.

  1. The meaning of the conclusions is unclear, and there are grammatical errors. The authors should think over the real significance of their results and try to rewrite this section to improve understanding of the conclusions.

Author Response

Responses to Reviewer 2 Comments

Manuscript ID: micromachines-1303474

Title: “Dynamic Performance of Partially Orifice Porous Aerostatic Thrust Bearing

The authors would like to thank the anonymous reviewers, the Editor-in-Chief and

the associate editor for their effort in reviewing the manuscript and for their valuable and constructive comments and fruitful observations that helped in improving the quality of the manuscript to a publishable standard. Detailed below are some responses to the reviewers’ comments and suggestions. Reviewers’ questions and comments are in BLACK and the authors’ answers and comments are in RED.

Point 1: There are numerous places in the text with English grammatical errors. The authors should be full checking for grammar and mistakes to meet the quality of Journal

Response 1: We appreciate your advice. To improve the English grammar and style of manuscript, a careful revision is made by the authors and also cross-verified with the help of native country speaker to improve the draft.  

Point 2: Most equations Tables in the paper, the authors don't mention to any references. Add citation to the equations which took from references.

Response 2: Relevant reference [29] for the reader’s review in more details is given on page 4-5. Table 1 and Table 2 are defined to identify the variables and design parameters.

Point 3: Add a new flowchart to explain the details of the selected approach.

Response 3: Flowchart is included to explain the details of the approach in methodologies section. Please refer the page number 8.

Point 4: Which software/ program that used as a tool to solve the equations of your problem? Give enough details about this point. You mention that authors used FLUENT to find the solution. But there are many equations included in the paper. Did authors used all these equations or just presented it.

Response 4: ANSYS workbench and 3D design software is used for the solution of the identified problem and for the simulation of the results.

Point 5: Did you check the accuracy of your numerical solution.

Response 5: Yes, the numerical solution results are checked with the simulated results to verify the results.

Point 6: How did you select the final numerical model? Based on what, explain this point.

Response 6: The model is based on improving  the performance.

Point 7: There are many research papers study the same problem that investigated in the present paper. What is exactly the new point of this work? The authors should focus to clarify this issue in the paper.

Response 7: Yes, in this work not only the performance is improved but also stability enhanced which imparts the reliability of the aerostatic bearing.

Point 8: The meaning of the conclusions is unclear, and there are grammatical errors. The authors should think over the real significance of their results and try to rewrite this section to improve understanding of the conclusions.

Response 8: Conclusion section is rewritten in order to bring required meaning and comprehensiveness.

The authors hope that the revised manuscript has satisfied the requirements from the reviewers, Associate Editor and the editor-in-chief. We would like to thank them for their very constructive comments, which are very insightful and beneficial for improving the quality of the manuscript.

Best regards,

Muhammad Punhal Sahto

Reviewer 3 Report

The authors simulated the dynamic stiffness and damping coefficients of aerostatic thrust bearings under different conditions. The simulated results are also verified via experiments. The authors concluded that the number of orifice, orifice diameter, orifice height, frequency will all affect the characteristics of bearings. I have some concerns regarding the content.

Major comments:

  1. Please clearly address the difference between this work and the cited references in chapter 1.
  2. Page 9. I can’t seem to understand how the authors can see the effect of different frequency in Figs. 3 and 4 (right before Ch4).
  3. In chapters 5 and 6, only the results are presented (what can be seen from the readers themselves), but no discussion on the underlying physics of the behaviors was included. This must be improved.

Minor comments:

  1. There are way too much typos, grammar errors, and missing words. Please double check the whole manuscript as well as improve the writing.
  2. Please place Fig. 1 to its correct position. It was placed in section 2.1 but was first discussed in section 2.2. Similar situation for Figs. 6 and 7. Figures 5 and 6 wasn’t even mentioned in their related sections. It seems like the manuscript was not prepared carefully.
  3. Please mark where the pressure distribution in Fig. 3b was obtained in Fig. 3a. Please also do the same for the rest of the figures.
  4. The contour plots are not shown with the same scale bar in Fig. 6. It is really hard for us to compare those cases with each other. Please revise that for all the contour plots in the manuscript.
  5. Figures 15 and 16 have the same figure caption. Again, I really feel the manuscript was not prepared carefully.
  6. Figure 17 was placed too far away from where it was mentioned in the content. This is not reader friendly.

Author Response

Responses to Reviewer 3 Comments

Manuscript ID: micromachines-1303474

Title: “Dynamic Performance of Partially Orifice Porous Aerostatic Thrust Bearing

The authors would like to thank the anonymous reviewers, the Editor-in-Chief and

the associate editor for their effort in reviewing the manuscript and for their valuable and constructive comments and fruitful observations that helped in improving the quality of the manuscript to a publishable standard. Detailed below are some responses to the reviewers’ comments and suggestions. Reviewers’ questions and comments are in BLACK and the authors’ answers and comments are in RED.

Point 1: Please clearly address the difference between this work and the cited references in chapter 1.

Response 1: The model used in this research work, which includes the porous and partial porous orifice restrictors whereas the cited literature review is only based on orifice and porous separately.

Point 2:  Page 9. I can’t seem to understand how the authors can see the effect of different frequency in Figs. 3 and 4 (right before Ch4).

Response 2: The description of Figures 3 and 4 are reset in the manuscript and also sequence of these Figures are changed and are mentioned in page number 9.

Point 3:  In chapters 5 and 6, only the results are presented (what can be seen from the readers themselves), but no discussion on the underlying physics of the behaviours was included. This must be improved.

Response 3: The results are presented along with analysis and are discussed for better clarity in results and analysis section.

Point 4: There are way too much typos, grammar errors, and missing words. Please double check the whole manuscript as well as improve the writing.

Response 4: We appreciate your advice. To improve the English grammar and style of manuscript, a careful revision is made by the authors and also cross-verified with the help of native country speaker to improve the draft.

Point 5: Please place Fig. 1 to its correct position. It was placed in section 2.1 but was first discussed in section 2.2. Similar situation for Figs. 6 and 7. Figures 5 and 6 wasn’t even mentioned in their related sections. It seems like the manuscript was not prepared carefully.

Response 5: Fig. 1,5,6, 7 and all other Figures are placed to its correct position in related sections.

Point 6:  Please mark where the pressure distribution in Fig. 3b was obtained in Fig. 3a. Please also do the same for the rest of the figures.

Response 6: The pressure distribution in Fig. 3a and 3b are provided and please refer the page number 9.

Point 7: The contour plots are not shown with the same scale bar in Fig. 6. It is really hard for us to compare those cases with each other. Please revise that for all the contour plots in the manuscript.

Response 7: The contour plots are revised in the whole manuscript.

Point 8: Figures 15 and 16 have the same figure caption. Figure 17 was placed too far away from where it was mentioned in the content. This is not reader friendly.

Response 8: Figures 15 and 16 captions are revised and space between Figure 16 and 17 are removed.

The authors hope that the revised manuscript has satisfied the requirements from the reviewers, Associate Editor and the editor-in-chief. We would like to thank them for their very constructive comments, which are very insightful and beneficial for improving the quality of the manuscript.

Best regards,

Muhammad Punhal Sahto

Round 2

Reviewer 1 Report

Accepted!

Author Response

Responses to Reviewer 1 Comments

Manuscript ID: micromachines-1303474

Title: “Dynamic Performance of Partially Orifice Porous Aerostatic Thrust Bearing

The authors would like to thank the anonymous reviewers and the associate editor for their effort in reviewing the manuscript and for their valuable and constructive comments and fruitful observations that helped in improving the quality of the manuscript to a publishable standard.

The authors hope that the revised manuscript has satisfied the requirements of the reviewers, Associate Editor, and the editor-in-chief. We would like to thank them for their very constructive comments, which are very insightful and beneficial for improving the quality of the manuscript.

I would thank to like to thank you for accepting the manuscript.

Best regards,

Muhammad Punhal Sahto

Reviewer 2 Report

  1. It should be redrawing the flowchart (Fig. 2) according to standard symbols. Because it was used incorrect symbols in the current flowchart.

Author Response

Responses to Reviewer 2 Comments

Manuscript ID: micromachines-1303474

Title: “Dynamic Performance of Partially Orifice Porous Aerostatic Thrust Bearing

The authors would like to thank the anonymous reviewers, the Editor-in-Chief and

the associate editor for their effort in reviewing the manuscript and for their valuable and constructive comments and fruitful observations that helped in improving the quality of the manuscript to a publishable standard. Detailed below are some responses to the reviewers’ comments and suggestions. Reviewers’ questions and comments are in BLACK and the authors’ answers and comments are in RED.

Point 1: Add a new flowchart to explain the details of the selected approach.

Response 3: Flowchart is revised as advised by author. Please refer the page number 8.

The authors hope that the revised manuscript has satisfied the requirements from the reviewers, Associate Editor and the editor-in-chief. We would like to thank them for their very constructive comments, which are very insightful and beneficial for improving the quality of the manuscript.

Best regards,

Muhammad Punhal Sahto

Reviewer 3 Report

Major comments:

  1. I’ve checked the revised chapters 5 and 6 (marked blue). I still cannot find any discussion on the underlying physics of the behaviors. What I meant was theories, mechanisms, equations that can support the correctness of the behaviors observed from those simulations must be included. Or simply put, whether if the results obtained are reasonable or not and why.

Minor comments:

  1. I mean please mark where is the line of pressure distribution shown in Fig. 4b (in the revised manuscript) located in Fig. 3a. The authors should draw a line in Fig. 4a to indicate where the pressure distribution in Fig. 4b is obtained. Please also do the same for the rest of the figures.
  2. The contour plots are still not shown with the same scale bar in in the revised manuscript except for Fig. 13. Please revise Figs. 7 and 10 accordingly. If I wasn’t clear enough, that means the pressure scale bars should have the same upper and lower limits in the same figure so readers can compare them.

Author Response

Responses to Reviewer 3 Comments

Manuscript ID: micromachines-1303474

Title: “Dynamic Performance of Partially Orifice Porous Aerostatic Thrust Bearing

The authors would like to thank the anonymous reviewers, the Editor-in-Chief and

the associate editor for their effort in reviewing the manuscript and for their valuable and constructive comments and fruitful observations that helped in improving the quality of the manuscript to a publishable standard. Detailed below are some responses to the reviewers’ comments and suggestions. Reviewers’ questions and comments are in BLACK and the authors’ answers and comments are in RED.

Point 1:  I’ve checked the revised chapters 5 and 6 (marked blue). I still cannot find any discussion on the underlying physics of the behaviours. What I meant was theories, mechanisms, equations that can support the correctness of the behaviours observed from those simulations must be included. Or simply put, whether if the results obtained are reasonable or not and why.

Response 1: The results are presented for better clarity in the results and analysis section.

Point 5: I mean please mark where is the line of pressure distribution shown in Fig. 4b (in the revised manuscript) located in Fig. 3a. The authors should draw a line in Fig. 4a to indicate where the pressure distribution in Fig. 4b is obtained. Please also do the same for the rest of the figures.

Response 2: The pressure distribution in Fig. 3a and 3b are provided and please refer the page number 9. The pressure distribution is marked for the rest of the figures in manuscript.

Point 2: The contour plots are still not shown with the same scale bar in in the revised manuscript except for Fig. 13. Please revise Figs. 7 and 10 accordingly. If I wasn’t clear enough, that means the pressure scale bars should have the same upper and lower limits in the same figure so readers can compare them.

Response 3: The contour plots are revised in the whole manuscript.

The authors hope that the revised manuscript has satisfied the requirements from the reviewers, Associate Editor and the editor-in-chief. We would like to thank them for their very constructive comments, which are very insightful and beneficial for improving the quality of the manuscript.

Best regards,

Muhammad Punhal Sahto
